# Identification of the LLDPE Constitutive Material Model for Energy Absorption in Impact Applications

**DOI:** 10.3390/polym13101537

**Published:** 2021-05-11

**Authors:** Luděk Hynčík, Petra Kochová, Jan Špička, Tomasz Bońkowski, Robert Cimrman, Sandra Kaňáková, Radek Kottner, Miloslav Pašek

**Affiliations:** 1New Technologies—Research Centre, University of West Bohemia, 301 00 Plzeň, Czech Republic; spicka@ntc.zcu.cz (J.Š.); tomasz@ntc.zcu.cz (T.B.); cimrman3@ntc.zcu.cz (R.C.); 2Faculty of Applied Sciences, University of West Bohemia, 301 00 Plzeň, Czech Republic; kochovap@kme.zcu.cz (P.K.); kanaksan@kme.zcu.cz (S.K.); radek.kottner@gmail.com (R.K.); 3MECAS ESI s.r.o., Brojova 2113, 326 00 Plzeň, Czech Republic; Miloslav.Pasek@esi-group.com

**Keywords:** LLDPE, quasi-static and dynamic experimental tests, impact energy absorption, material parameter identification, constitutive material model, validation, simulation

## Abstract

Current industrial trends bring new challenges in energy absorbing systems. Polymer materials as the traditional packaging materials seem to be promising due to their low weight, structure, and production price. Based on the review, the linear low-density polyethylene (LLDPE) material was identified as the most promising material for absorbing impact energy. The current paper addresses the identification of the material parameters and the development of a constitutive material model to be used in future designs by virtual prototyping. The paper deals with the experimental measurement of the stress-strain relations of linear low-density polyethylene under static and dynamic loading. The quasi-static measurement was realized in two perpendicular principal directions and was supplemented by a test measurement in the 45° direction, i.e., exactly between the principal directions. The quasi-static stress-strain curves were analyzed as an initial step for dynamic strain rate-dependent material behavior. The dynamic response was tested in a drop tower using a spherical impactor hitting a flat material multi-layered specimen at two different energy levels. The strain rate-dependent material model was identified by optimizing the static material response obtained in the dynamic experiments. The material model was validated by the virtual reconstruction of the experiments and by comparing the numerical results to the experimental ones.

## 1. Introduction

Thin-layered polymer materials are traditionally used for packaging goods to protect them during transportation. Therefore, the major desired properties relate to thickness, density (which relates to weight), strength, elongation, puncture resistance, and stretching level; see Table 1. On the other hand, preliminary experimental tests also show the good performance of such materials in energy absorption.

Current trends in the automotive industry regarding future mobility bring new challenges for energy-absorbing safety systems. Non-traditional seating configurations in autonomous vehicles and complex crash scenarios including multi-directional loading are to be considered hand-in-hand with advanced materials for energy absorption. The study [1] used a numerical simulation approach to assess the newly patented safety system (see Figure 1) [2]. The system is based on two layers of a multi-layered membrane injected from the roof between the windshieldand the front seats, catching the driver and the passenger during an accident in a similar manner as an airbag performs. The advantage of the approach over the airbag is the simple implementation for multi-directional impact loading and addressing the out-of-position seating issue.

As virtual prototyping plays an important role currently in the design of new products, the paper aimed to identify the parameters of the linear low-density polyethylene (LLDPE) material for both static and dynamic loading, to implement them in a constitutive material model, and to verify the material model by numerical simulations representing the experiments. As the static tests were represented by quasi-static loading conditions, the dynamic tests represented a scenario close to the one schematically described in Figure 1.

LLDPE films have been identified as the most promising material in cases where impact loading is assumed, because of their higher average peak force and the energy-to-peak force when compared to LDPE [3]. LLDPE is a linear polyethylene with a significant number of short branches (see Figure 2) commonly made by copolymerization of ethylene and another longer olefin, which is incorporated to improve properties such as tensile strength or resistance to harsh environments. The structure of LLDPE leads to its heterogeneous non-linear behavior.

LLDPE is very flexible, elongates under stress, absorbs a high level of impact energy, and thus, is suitable for making thin and ultra-thin films [5,6,7,8]. The mechanical properties of polyethylene depend on its complex structure [9], which leads to non-linear heterogeneous behavior during mechanical and numerical tests. This behavior has been explained by other authors, e.g., [4,10,11], where the differences in the chain structures among HDPE (high-density polyethylene), LLDPE, and LDPE (low-density polyethylene) are described. The LLDPE film MD tear strength is dependent on the utilized comonomers (higher for hexene- and octene-based resins whilst lower for butene-based resins) [12] with the LLDPE Poisson ratio equal to ν=0.44 for LLDPE [13,14].

The main mechanical characteristics of polyethylene are the yield stress and the yield strain, corresponding to the point where plastic non-recoverable deformation due to permanent changes in the polymer chains starts. The yield stress and the yield strain of LLDPE depend on the temperature and the strain rate [5,6,15]. The yield stress increases while the yield strain decreases with rising strain rate [9]. The double yield point is also mentioned in the literature [16]. The relation between yield stress, temperature, and strain rate can be described by constitutive laws [5,6,9,17], and the temperature-dependent mechanical properties of thin-layered materials have been addressed [18]. Upon comparing LDPE, LLDPE, and HDPE, LLDPE showed greater rate sensitivity than the other two materials under both static and dynamic regions of a compression test [9].

The typical stress-strain relation, as well as the strain rate dependence are drawn in Figure 3. The tensile properties are dependent on the strain rate [19], and also, the yield stress depends on the strain rate [20].

The typical stress-strain curves of LLDPE begin by an initial elastic Region I, followed by yielding that is accompanied by neck propagation in Region II; see Figure 3. The third Region III is the stiffening leading to material rupture [21].

LLDPE has an anisotropic behavior due to its chain structure. The chain structure creates the anisotropy in two perpendicular directions, called the machine direction (MD) and the transversal direction (TD). The local preferential orientation of chains in LLDPE affects the tensile strength in the MD and TD [11]. In the direction of the main chain orientation, mostly the MD, LLDPE is stiffer than in the perpendicular direction, mostly the TD [3,5,6,15]. The tensile stress-strain relations in the MD and TD play an important role during the biaxial deformation of the impact test [3].

## 2. Materials and Methods

Material parameter identification was applied to the commercial LLDPE thin foil [22]. The selected foil is a commonly produced foil supplied in rolls with a width of 100–500 mm and a thickness of 4–50 µm. It was selected due to its common production and low price. Table 1 summarizes its parameters presented by the producer.
polymers-13-01537-t001_Table 1Table 1LLDPE properties [22].Physical PropertiesUnitTolerance ±ValueTesting MethodThicknessµm212Thickness gaugeWidthmm5500Measuring tapeLength
-5High-speed encoderDensityg/cm3-0.91–0.92ASTM D-1505 [23]**Mechanical Properties****Unit****Tolerance**±**Value****Testing Method**Tensile strength MDMPa1029.2ASTM D-882 [23]Tensile strength TD
14.1
Break elongation MD%245
Break elongation TD
540
Dart dropg40ASTM D-1709 [23]Puncturekg1.7High-light testerStretching level--110



### 2.1. Quasi-Static Loading

The unilateral quasi-static loading test of the material sample was executed using a 574LE2 TestResources testing machine. From the material roll (see Figure 4) provided by the producer, testing samples of length $l_{0}=5$ mm and width w=10 mm were extracted; see Figure 5a, where the left and right yellow sides are fixed to the testing machine jaws. The thickness of the sample was h=12 µm. The samples were fixed in the testing machine jaws (see Figure 5b) and stretched in two major orthotropic directions (MD and TD). The MD is in the direction in which the material is wound up on the roll, whist the TD is perpendicular to the MD; see Figure 4.

Several samples were tested in each direction at three different stretching velocities *v*, namely 0.0002, 0.02, and 0.2 m/s, half per each side of the jaws. Complementary tests in the directions between the MD and TD (labeled as D3 and D4; see Figure 4) were done to check the influence of fiber direction on the material behavior in the skewed (45${}^{{}^{\circ}}$) direction. Table 2 summarizes all the quasi-static tests. N=6 samples were measured in each direction for each velocity except v=0.2 m/s, where D4 did not need to be measured. As the additional measurements in D3 showed a consistent skewed behavior in all three velocities and the additional measurement in D4 (which is D3 just rotated around 90${}^{{}^{\circ}}$) confirmed the skewed behavior for the first two velocities, the measurement for the last velocity was performed only in MD, TD, and D3. The particular test finished when the sample ruptured.

During the sample stretching, force *F* versus displacement *d* was recorded. Based on the sample size with the sample initial length $l_{0}$ and initial cross-sectional area $A_{0}=hw$, the engineering stress σ versus engineering strain ϵ curves were calculated as:(1)$$\sigma =\frac{F}{A_{0}},\;\epsilon =\frac{d}{l_{0}}.$$

The constant Young modulus *E* was also identified as the slope of the initial elastic region as:(2)$$\sigma =\frac{F}{A_{0}}=E\frac{d}{l_{0}}\;\Rightarrow \;E=\frac{Fl_{0}}{dA_{0}}.$$

Fulfilling the aim of this study, the quasi-static tests were reproduced by the numerical simulations. The simulation was realized in Virtual Performance Solution (VPS by ESI Group), Version 2020. Following the structure of LLDPE in Figure 3 (2 mutually perpendicular sets of fibers), the material model 151 Fabric Membrane Element with Nonlinear Fibers [24] from the ESI constitutive material model database was proposed. According to membrane theory, the resultant stress curves were calculated by multiplying the engineering stress by the membrane thickness as:(3)$$\sigma_{h}=\sigma h$$
in both the MD and TD. The resulting material curves taken as the average curves from the quasi-static test measurements in particular directions served as the constitutive data to feed the material model 151. The model concerned 2 sets of fibers, whose stress versus strain relation was defined by the resultant engineering stress versus engineering strain curve. The angle between the sets of fibers was 90${}^{{}^{\circ}}$. The resultant shear stress necessary to complete the membrane material model was calculated using the measurement in direction D3, as shown in Figure 6.

Supposing a square sample, the shear force *Q* and shear angle γ were calculated through the following formulas.
(4)$$Q=\frac{F_{3}}{2\cos\frac{\psi }{2}},$$
where the shear angle:(5)$$\gamma =\frac{\pi }{2}-\psi$$
was calculated based on the deformed sample angle ψ as:(6)$$\cos\frac{\psi }{2}=\frac{\sqrt{2}L+d}{2L},$$
where *L* is the side of the square sample, *d* is the displacement in direction D3, and $F_{3}$ is the force recorded in direction D3. Therefore, the shear stress can be calculated as:(7)$$\tau =\frac{Q}{Lh}$$
and the resultant shear stress is:(8)$$\tau_{h}=\tau h=\frac{Q}{L}.$$

The thickness of the material was h=12 µm, as defined by the producer [22]. In addition to the stress-strain constitutive relations, the chosen material model [24] requires also the amount of energy absorption. The energy absorption was calculated from the dynamic experimental measurements, and other numerical parameters feeding the material model were used as proposed by the VPS manual [24].

The quasi-static numerical test was reconstructed to confirm the chosen material model. A single 4-node membrane element model was loaded by stretching both sides of the element by the 3 different loading velocities *v*, namely 0.0002, 0.02, and 0.2 m/s, half per each side of the jaws; see Figure 7.

The element section force leading to the resultant stress was recorded during the simulation to be compared to the experimental data.

### 2.2. Dynamic Loading

The dynamic tests, carried out to reproduce the scenario from Figure 1, took the form of drop tests of a spherical impactor falling at a given velocity on a multi-layered material sample. A special drop tower was designed for this purpose; see Figure 8.

According to Figure 1, the drop test was used to simulate a collision scenario similar to the impact of a human head into the safety layers during a frontal crash. Typical impacts for testing safety systems are designed for velocities $v_{0}$ equal to 30 and 50 km/h [25], corresponding to those used in sled tests. As the mass of the human head is approximately m=4.5 kg [26] and the mass of the testing impactor is M=10.72 kg, the drop test height *H* was calculated from the energy balance equation:(9)$$\frac{1}{2}mv_{0}^{2}=MgH$$
using gravity acceleration g=9.81 m/s${}^{2}$. Equation (Equation 9) yields drop heights equal to 1.49 and 4.13 m for the velocities 30 and 50 km/h, respectively. Due to the design limitations (limited maximum height of the drop tower, which was also a limitation of this study), only height H=1.5 m corresponding to velocity $v_{0}=30$ km/h was considered. To include different impact velocities to optimize the constitutive material model, additional tests at height H=1 m corresponding to velocity $v_{0}=25$ km/h were carried out. Relating the energy balance with the head impactor mass M=10.72 kg, the impact velocities corresponded to 4.43 and 5.43 m/s for H=1 and 1.5 m, respectively.

As the dynamic impact loading was aggressive, the target material was wound onto the frame in several layers; see Figure 9. Preliminary experiments showed a sufficient number of layers *n* to be 8, 9, and 10, so the matrix of experiments contained 2 drop heights (10 and 15 dm) × 3 sets of layers (8, 9, and 10).

Table 3, summarizing the drop tests, shows that in the end, only 5 experimental drop tests were used for the optimization procedure, as the most aggressive one, meaning the fall from the greatest height H=15 dm onto the lowest number of layers n=8, ruptured the target material layers. The last column of Table 3 designates the identification of the particular drop tests in the following figures and analyses.

The acceleration was measured using a Kistler 8742A5 uniaxial piezoelectric accelerometer fixed to the impactor, with the axis of the measurement parallel to the axis of the impactor. The impactor was held by an electromagnet, and the free-fall motion was controlled by a linear guide; see Figure 9. Additionally, the deflection of the impactor was measured with a Micro-Epsilon optoNCDT 2300-50 laser measuring system connected to an NI 9214 voltage input module in the NI cDAQ-9178 chassis. The final time-correlated signals were recorded by NI Signal Express Software. The measured acceleration signal was filtered by the CFC 1000 filter [27]. From the physical principle, the piezoelectric accelerometer cannot measure free-fall gravity acceleration [28]. The experimental acceleration curve decreased to minus g just after release and reached the equilibrium of 0 g during the free fall, so the experimental acceleration curve needed to be adjusted to be comparable to the simulation results.

As the measured displacement was limited by the range of the laser measuring system, double integration of the acceleration signal was used to extend the displacement in the whole time interval of the loading and unloading phases of the impact. Using the updated acceleration and displacement signals, the total energy of the system as the sum of the kinetic energy, the potential energy, and the work done at any time, respectively, were monitored as:(10)$$E\left(t\right)=E_{k}+E_{p}+W=\frac{1}{2}Mv{\left(t\right)}^{2}+Mgd\left(t\right)+\int_{0}^{d(t)}Ma\left(t\right)ds$$
to check the correctness of the calculations as it must be constant during the action. Here, a(t) is the updated measured time-dependent impactor acceleration, and the impactor velocity v(t) and the impactor displacements d(t) were calculated by the first and the second integration, respectively, of the acceleration signal a(t). The gravity acceleration g was subtracted from the impactor acceleration to subtract the work done by the potential energy.

Marking $E_{kp}\left(t\right)=E_{k}\left(t\right)+E_{p}\left(t\right)$ as the sum of the kinetic energy and potential energy at any time, the energy absorption was calculated as the energy loss at any time:(11)$$D=1-\frac{\Delta E_{u}}{\Delta E_{l}},$$
where $\Delta E_{l}=\max E_{kp}\left(t\right)-\min E_{kp}\left(t\right){|}_{\mathrm{loading}}$ is the difference between the maxima of the sum of the kinetic energy and potential energy during the loading phase and $\Delta E_{u}=\max E_{kp}\left(t\right)-\min E_{kp}\left(t\right){|}_{\mathrm{unloading}}$ is the energy difference between the maxima of the sum of the kinetic energy and potential energy during the unloading phase, where the resting energy is absorbed by the material work in order to have the constant total energy E(t) from Equation (Equation 10).

### 2.3. Identification of Dynamic Material Parameters

As the material properties of LLDPE are strain rate dependent [21], the constitutive material curves achieved by the quasi-static experimental measurements were used as the initial optimization step for the optimization of the dynamic material parameters. The optimization was done using the numerical simulation reproducing the drop test experiment. The strain rate-dependent curves from the first optimization (H=10 dm and n=8 layers) were used as the initial curves for the other optimization runs to speed up the optimization process.

The standard MATLAB function fminsearch was adopted to optimize the values for the stiffness and the yield stress in the two directions MD and TD towards the expected values. According to Figure 3, the stiffness of Region I and the yield stress were optimized. For the optimization purposes, Region I was divided into 2 strain intervals $\epsilon \in \left[0,\epsilon_{y1}^{\mathrm{MD}}\right]\cup \left(\epsilon_{y1}^{\mathrm{MD}},\epsilon_{y2}^{\mathrm{MD}}\right]$ for the MD response and $\epsilon \in \left[0,\epsilon_{y1}^{\mathrm{TD}}\right]\cup \left(\epsilon_{y1}^{\mathrm{TD}},\epsilon_{y2}^{\mathrm{TD}}\right]$ for the TD response. The corresponding stress interval was composed as $\sigma \in \left[0,\sigma_{y1}^{\mathrm{MD}}\right]\cup \left(\sigma_{y1}^{\mathrm{MD}},\sigma_{y2}^{\mathrm{MD}}\right]$ for the MD response and $\sigma \in \left[0,\sigma_{y1}^{\mathrm{TD}}\right]\cup \left(\sigma_{y1}^{\mathrm{TD}},\sigma_{y2}^{\mathrm{TD}}\right]$ for the TD response so that the yield points $\left[\epsilon_{y1}^{\mathrm{MD}},\sigma_{y1}^{\mathrm{MD}}\right]$ and $\left[\epsilon_{y2}^{\mathrm{MD}},\sigma_{y2}^{\mathrm{MD}}\right]$ in the MD response and $\left[\epsilon_{y1}^{\mathrm{TD}},\sigma_{y1}^{\mathrm{TD}}\right]$ and $\left[\epsilon_{y2}^{\mathrm{TD}},\sigma_{y2}^{\mathrm{TD}}\right]$ in the TD response were introduced. Addressing the resultant stress in Equation (Equation 3), the MD and TD curves in Region I were updated as: (12)$$\begin{array}{ccc} \sigma_{h}^{\mathrm{MD}}:=\sigma_{h}^{\mathrm{MD}}\left(\epsilon ,k_{1},k_{e},k_{y}\right) & = & k_{y}\sigma_{h}^{\mathrm{MD}}\left(\frac{\epsilon }{k}\right),\;k=\left\{\begin{array}{cc} k_{1}k_{e} & \forall \epsilon \in [0,\epsilon_{y1}^{\mathrm{MD}}]\\ k_{e} & \forall \epsilon \in (\epsilon_{y1}^{\mathrm{MD}},\epsilon_{y2}^{\mathrm{MD}}] \end{array}\right. \end{array}$$(13)$$\begin{array}{ccc} \sigma_{h}^{\mathrm{TD}}:=\sigma_{h}^{\mathrm{TD}}\left(\epsilon ,k_{1},k_{e},k_{y}\right) & = & k_{y}\sigma_{h}^{\mathrm{TD}}\left(\frac{\epsilon }{k}\right),\;k=\left\{\begin{array}{cc} k_{1}k_{e} & \forall \epsilon \in [0,\epsilon_{y1}^{\mathrm{TD}}]\\ k_{e} & \forall \epsilon \in (\epsilon_{y1}^{\mathrm{TD}},\epsilon_{y2}^{\mathrm{TD}}] \end{array}\right. \end{array}$$
by multiplying by dimensionless coefficients $\frac{1}{k_{1}k_{e}}$, $\frac{1}{k_{e}}$, and $k_{y}$ during the optimization process. The coefficient *k* scales with the strain in Region I, in particular $k_{1}k_{e}$ until the first yield point is reached and $k_{e}$ further between both yield points. The coefficient $k_{y}$ scales with the resultant stress. Such a parametric representation of the constitutive curves was proposed based on the preliminary numerical tests, which also confirmed the use of the same multipliers $k_{1}$, $k_{e}$, and $k_{y}$ for both Equations (Equation 12) and (13) to hold the physical meaning of the optimized constitutive curves. For the independent sets of coefficients for the curves in the MD and TD, the optimizer strengthened the MD, whilst the TD was completely suppressed. Therefore, both sets of coefficients needed to be constrained together. The optimization process was run in a loop controlled by a MATLAB script updating the constitutive material curves in the MD and TD according to Equations (Equation 12) and (13). The cost function in the optimization measured the relative acceleration error $E_{a}$ defined as:(14)$$E_{a}=\frac{∥a_{s}\left(t\right)-a_{e}\left(t\right)∥}{∥a_{e}\left(t\right)∥}{|}_{t\in [t_{1},t_{m}]},$$
where $a_{e}\left(t\right)$ is the time-dependent acceleration signal measured from the experiment, $a_{s}\left(t\right)$ is the time-dependent acceleration response calculated by the numerical simulation, and *t* is the time in the error calculation interval $\left[t_{1},t_{m}\right]$. As well as the experimental acceleration signal, the calculated acceleration signal was also filtered by the CFC 1000 filter [27]. Figure 10 shows the simulation setup for the optimization runs. The initial pre-strain of the material wound on the frame was estimated based on preliminary numerical simulations to be 10%, i.e., $\epsilon_{0}$ = 0.1 in the MD. The displacement error $E_{d}$ was calculated similarly to the acceleration error as:(15)$$E_{d}=\frac{∥d_{s}\left(t\right)-d_{e}\left(t\right)∥}{∥d_{e}\left(t\right)∥}{|}_{t\in [t_{1},t_{m}]},$$
where $d_{e}\left(t\right)$ is the time-dependent displacement signal obtained by the double integration of the acceleration signal and $d_{s}\left(y\right)$ is the time-dependent displacement response calculated by the numerical simulation.

The interval for calculating the acceleration error in Equation (Equation 14) was limited to the loading phase for $t\in [t_{1},t_{m}]$ because the constitutive material model was developed for the energy absorption during the stretching. Moreover, expanding the time interval to the unloading phase negatively influenced the optimized curve fit during the loading phase. The discretization of the time interval as $t\in \{t_{1},\cdots ,t_{i},\cdots ,t_{m}\}$ led to the cost function:(16)$$f=\frac{\sum_{i=1}^{m}\left[a_{s}\left(t_{i}\right)-a_{e}\left(t_{i}\right)\right]}{\sum_{i=1}^{m}a_{e}\left(t_{i}\right)},\;i\in \left\{1,2,\cdots ,m\right\},$$
where $a_{e}\left(t_{i}\right)$ is the measured acceleration signal sampled at discrete times $t_{i}$ and $a_{s}\left(t_{i}\right)$ is the calculated acceleration signal based on the constitutive curves from Equations (Equation 12) and (13). Therefore, the cost function had the form:(17)$$f=f(k_{1},k_{e},k_{y}),$$
depending on three coefficients, $k_{1}$, $k_{e}$, and $k_{y}$, whose values were updated during the optimization process by the standard MATLAB function fminsearch. The update of the quasi-static constitutive curves is illustrated in Figure 11. Note that evaluating the three-parameter function *f* in Equation (Equation 17) involved running a finite element simulation of the drop test to get $a_{s}\left(t_{i}\right)$. Considering that ϵ and $\sigma_{h}$ represent the strain and the resultant stress, respectively, the optimization loop was:
Update both the MD and TD curves according to Equations (Equation 12) and (13):
(a)$\forall \epsilon \in [0,\epsilon_{y_{1}}]$ update the stiffness by changing the slopes of the curves using $\epsilon :=\frac{1}{k_{1}k_{e}}\epsilon$;(b)$\forall \epsilon \in (\epsilon_{y_{1}},\epsilon_{y_{2}}]$ update the stiffness by changing the slopes of the curves using $\epsilon :=\frac{1}{k_{e}}\left(\epsilon -\epsilon_{y_{1}}\right)+\epsilon_{y_{1}}$;(c)$\forall \epsilon \in [0,\epsilon_{y_{2}}]$ update the resultant stress as $\sigma_{h}:=k_{y}\sigma_{h}$;(d)$\forall \epsilon >\epsilon_{y_{2}}$ connect the parts of the curves in Regions II and III to the second yield point using $\epsilon :=\epsilon +\Delta \epsilon_{y_{2}}$ and $\sigma_{:}=h\sigma_{h}+\Delta \sigma_{hy_{2}}$ where $\left[\Delta \epsilon_{y_{2}},\Delta \sigma_{{hy}_{2}}\right]$ is the shift of the second yield point;Run the VPS simulation to get $a_{s}\left(t_{i}\right)$ for i∈{1,2,⋯,m};Evaluate the cost function *f* in Equation (Equation 16);Repeat the loop from 1 until the cost function *f* reaches its minimum;Return both the MD and TD curves according to Equations (Equation 12) and (13) for the optimized coefficients, $k_{1}$, $k_{e}$, and $k_{y}$.

The optimization loop is illustrated in Appendix A as a flowchart.

For each testing scenario with *n* layers, the material was modeled by single-layered membrane elements, where the number of upper and lower layers of the model was specified using the membrane material thickness defined by multiplying the single-layer thickness *h* by the number of layers *n*, meaning that the resultant stress curves in Equations (Equation 3) and (Equation 8) were also multiplied by *n* for the particular model. Both sides of the layers were fixed by boundary conditions representing the attachment to the frame. The spherical impactor was modeled as a rigid body situated just above the upper layer and loaded by the initial velocity *v* corresponding to the particular height. The vertical acceleration and the vertical displacement were stored and compared to the experimental data.

## 3. Results

All equations stated in the paper are summarized in Appendix B. The following figures and tables summarize the results from the quasi-static tests, as well as the identification of LLDPE parameters under dynamic loading.

### 3.1. Quasi-Static Loading

The quasi-static experiments proved that the typical stress versus strain curve for LLDPE was composed of three regions [21]; see Figure 3. A summary of all results obtained by static experimental measurements under different quasi-static loading velocities using a single material layer is displayed in Figure 12. The curves are cut at the positions of the sample ruptures. Table 4 compares the measured experimental properties to those defined by the producer [22].

As the quasi-static tests in all three stretching velocities showed similar performance, the curves for each direction were averaged—as shown in Figure 13. It can be seen that the stretching responses in the skewed directions D3 and D4 fit between the MD and TD curves, so no unpredictable behavior during the multi-directional loading should be expected. Therefore, the skewed direction D3 was also used to identify the shear behavior according to Equations (Equation 4)–(Equation 8).

Whilst Figure 13a shows the force dependent on the displacement averaged per the direction and per the stretching velocity, Figure 13b shows the total average of the calculated stress versus strain curves in the MD and TD calculated using Equation (Equation 1) for each quasi-static test measurement. In Figure 13b, points A and B represent the yield points in the MD, and points C and D represent the yield points in the TD. Equation (Equation 1) relates F(d) and σ(ϵ) between Figure 13a,b.

By the detailed analysis of the measured data in Figure 13b, the double yield point [16] from Equations (Equation 12) and (13) was observed in both directions. In the MD, the first point appeared at the stress of $\sigma_{y1}^{\mathrm{MD}}=8.4$ MPa, which corresponded to the strain of $\epsilon_{y1}^{\mathrm{MD}}=0.26$. The second yield point appeared by reaching the stress of $\sigma_{y2}^{\mathrm{MD}}=20$ MPa, which corresponded to the strain of $\epsilon_{y2}^{\mathrm{MD}}=0.84$. In TD, the first yield point appeared at the stress of $\sigma_{y1}^{\mathrm{TD}}=8$ MPa, which corresponded to the strain of $\epsilon_{y1}^{\mathrm{TD}}=0.33$. The second yield point appeared before reaching the maximum stress in Region I at the stress of $\sigma_{y2}^{\mathrm{TD}}=10$ MPa corresponding to the strain of $\epsilon_{y2}^{\mathrm{TD}}=0.69$. Table 5 summarizes the yield points.

Taking into account the elastic region, the Young modulus E=50 MPa was identified using Equation (Equation 2) by averaging the slopes of the elastic regions of all curves; see Table 6. The average was calculated for the particular directions and stretching velocities firstly leading to the global average. Both the MD and TD were averaged as they exhibited similar stiffness in the first region.

Finally, the resultant constitutive material stress curves developed using Equations (Equation 3) and (Equation 8) for a single layer of LLDPE were calculated for the quasi-static loading to feed the constitutive material model; see Figure 22. A single-element numerical simulation to reproduce the stretching was run. Figure 14 shows a perfect fit to the experimental curves.

### 3.2. Dynamic Loading

The acceleration decrease interval from 0 g to minus g within approximately the first 32 ms was used as the approximated parabolic acceleration ramp (see Figure 15) after the first contact of the impactor, where the mirrored signal from minus g to 0 g was added to the measured acceleration in the first 50 ms after the first contact between the impactor and the material; see Figure 16, Figure 17, Figure 18, Figure 19 and Figure 20. By this, the inability of the acceleration sensor to measure the free fall acceleration was mitigated.

The time of the first contact of the impactor with the material, as well as the impact velocity were estimated from the ideal free fall from the height *H* after releasing the electromagnet. Due to uncertainty in the frame versus impactor linear guide friction, the related actual impact velocity, and time of contact, an iterative process starting from the free fall assumptions was used to determine the actual moment of impact and the impact velocity, based on comparing the doubly integrated accelerations to the displacements obtained by the laser measuring system.

Such a process led to a perfect fit in both measured and calculated displacements (both shown in Figure 16, Figure 17, Figure 18, Figure 19 and Figure 20) identifying also the real impact velocity (see Table 7). The only exception was Scenario 1509, where the displacement measurement failed. Therefore, the impact velocity was estimated to fit the remaining part of the displacement curve.

The dynamic loading proved the strain rate dependency of LLDPE. LLDPE also exhibited strong energy absorption. The energy absorption was calculated by Equation (Equation 11) and was identified as being similar for all five drop test scenarios and averaged per drop height to obtain the final average D=88.96% (see Table 7) used for the constitutive material model.

### 3.3. Identification of the Dynamic Material Parameters

Several approaches to optimize the strain rate-dependent constitutive material curves were used, and in the end, the same stiffening ratio in the MD and TD was proposed. Equations (Equation 12) and (13) were designed to describe the stress-strain relation as a result of the preliminary optimization tests. The numerical tests showed that a purely linear response in Region I did not fit the experimental data sufficiently. Therefore, an additional constant $k_{1}$ was introduced to make Region I partially linear. The optimization process controlled by a MATLAB script involved running a series of simulations for updating the constitutive material model curves. The quasi-static response was taken as the initial guess for the optimization.

Table 8 shows the coefficients coming from the optimization process. Table 8 also shows the number of iterations leading to the optimized constitutive material curves, as well as the errors from the cost function calculated by Equation (Equation 14) and the error in the displacement calculated by Equation (Equation 15).

The intervals for calculating acceleration error are delimited in Figure 16, Figure 17, Figure 18, Figure 19 and Figure 20 by red dotted vertical lines to consider only the loading, where the iterative processes for the particular drop heights and the particular number of layers are shown.

The original experimental curves are in red dashed lines. The updated target curves (displacement obtained by integration and acceleration updated by gravity) are shown in dashed black lines. The initial curves (using the static constitutive material model) for optimization iterations are shown in dashed blue lines. The optimized curves are shown in solid blue lines. The iterative process is shown in solid grey curves.

All the identified strain rate-dependent engineering stress versus engineering strain constitutive material curves in both the MD and TD are shown in Figure 21a. Due to the two different drop heights and three different sets of multiple layers, each drop scenario provided a different strain rate-dependent response, so all tests were normalized by the number of layers, which led to similar strain rate constitutive material curves for a single layer in both the MD and TD; see Figure 21b.

As the difference between the curves corresponded to the difference during the experimental measurement, the constitutive material curves in the MD and TD were identified by averaging the drop tests; see Figure 22a. Figure 22b shows the shear stress versus shear strain as calculated by Equations (Equation 3)–(Equation 8) for a single layer.

Using the identified averaged constitutive material curves in both the MD and TD and the average energy absorption, all drop tests were reconstructed by numerical simulations. The results are shown in Figure 23, Figure 24, Figure 25, Figure 26 and Figure 27.

Table 9 shows the agreement in acceleration and displacement for all the drop tests using the averaged constitutive material curves.

Figure 23 compares the simulation to the experimental drop test for the drop height H=10 dm and the number of layers n=8. Figure 24 compares the simulation to the experimental drop test for the drop height H=10 dm and the number of layers n=9. Figure 25 compares the simulation to the experimental drop test for the drop height H=10 dm and the number of layers n=10. Figure 26 compares the simulation to the experimental drop test for the drop height H=15 dm and the number of layers n=9. Figure 27 compares the simulation to the experimental drop test for the drop height H=15 dm and the number of layers n=10.

## 4. Discussion

The quasi-static experiments were performed in two perpendicular directions supported by measurements in two skewed directions. Although the MD and TD exhibited different loading behavior, the measurements in the skewed directions supported the fact that there was no unexpected behavior during loading in any auxiliary direction.

Table 4 shows a good agreement with the factory data of the quasi-static experimental test regarding the tensile stress in both directions. The break elongation was 30% higher in the TD and 43% lower in the MD when compared to the material data sheet in Table 1, which might be caused by the laboratory conditions and influenced by the specimen size. The experimental measurements also confirmed previous studies showing that LLDPE is stiffer in the MD compared to the TD [3,5,6,15].

Table 5 summarizes the yield stresses $\sigma_{y}^{\mathrm{MD}}=8.4$ MPa and $\sigma_{y}^{\mathrm{TD}}=8$ MPa, as well as the yield strains $\epsilon_{y}^{\mathrm{MD}}=0.26$ and $\epsilon_{y}^{\mathrm{TD}}=0.33$, which were comparable to the values presented in the literature [21], where the yield stress $\sigma_{y}=9.9$ MPa and the yield strain ϵ=0.33. However, the elongation at break was measured equal to 1045%, which was higher than those measured and stated by the material data sheet. The Young modulus in Table 6
E= 50 MPa also showed a comparable value to the published values [21], where the Young modulus was experimentally identified as E=64 MPa.

The drop test experimental measurements proved the considerable energy absorption summarized in Table 7, which was used in the constitutive material model for the dynamic response. To maintain a stable optimization in the MD and TD, the same multipliers were supposed for developing the dynamic constitutive material model in the MD and TD. The optimized multipliers, as well as the optimization process errors are stated in Table 8. The optimization process led to stiffening of about 3.5 times for the drop height H=10 dm as the stiffening is about 2 times for the drop height H=15 dm. The yield stress balanced around the measured quasi-static value.

The acceleration error was calculated only during the loading phase, because of the complex unloading behavior and because of the fact that the constitutive material model was developed for the energy absorption during the loading.

The dynamic response exhibited similar values for both drop heights, so single dynamic constitutive material curves were developed by averaging the particular response curves in the MD and TD. The averaged constitutive material curves in the MD and TD were then used to recalculate all the drop tests again with the error shown in Table 9. The developed constitutive material model described the LLDPE film behavior to be used for energy absorption during the impact well.

Even though the identified constitutive material model described the expected scenario for the energy absorption, future work will consider the identification of dynamic constitutive material curves for different loading patterns and different drop energy, which was also a limitation of the current study. The study was also limited by the height of the drop test tower to address only the lower velocity levels. Therefore, future development would enable the use of the constitutive material model to be implemented for a wider spectra of impact scenarios with energy absorption.

## 5. Conclusions

The paper contributed to the field of virtual testing by developing a material model and identifying its constitutive parameters. The target material was LLDPE, a material traditionally used for packaging goods to protect them during transportation. The paper proved the high energy absorption of the material suitable for impact protection, also due to its low weight. Both the quasi-static and dynamic responses of the material were considered in the constitutive material model.

Besides the identification of the constitutive material parameter for both the quasi-static and dynamic responses, the paper provided a complex description of the experimental measurements. While the quasi-static response was measured using a unilateral stretch measurement in the MD and TD, the dynamic tests employed a sphere impact using a drop tower.

The quasi-static response was analyzed and evaluated based on the measurement of several samples providing the final curves describing the resultant stress dependent on the strain in the MD and TD. Those quasi-static curves served as initial values for the dynamic response, which was optimized by aligning the experimental and calculated accelerations of the impactor.

A good agreement of the experimental and model results was achieved and reported, providing the linear low-density polyethylene material model for virtual testing.

## Figures and Tables

**Figure 1 polymers-13-01537-f001:**
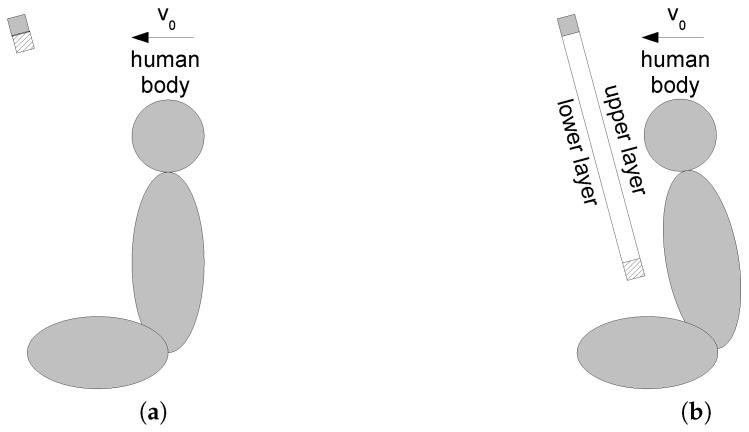
Scheme of a new safety system for absorbing impact energy: (**a**) Folded. (**b**) Unfolded.

**Figure 2 polymers-13-01537-f002:**
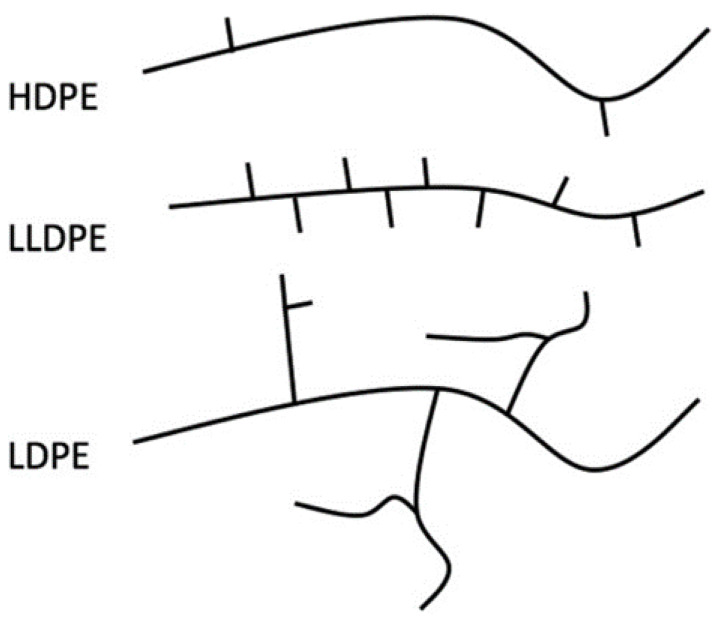
Chain structures of HDPE, LLDPE, and LDPE [4].

**Figure 3 polymers-13-01537-f003:**
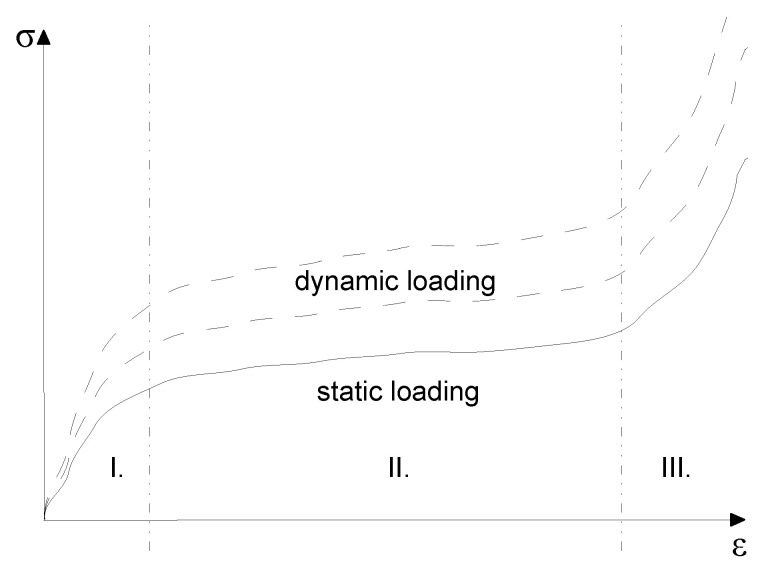
Typical stress-strain curve of LLDPE with different types of loading [21].

**Figure 4 polymers-13-01537-f004:**
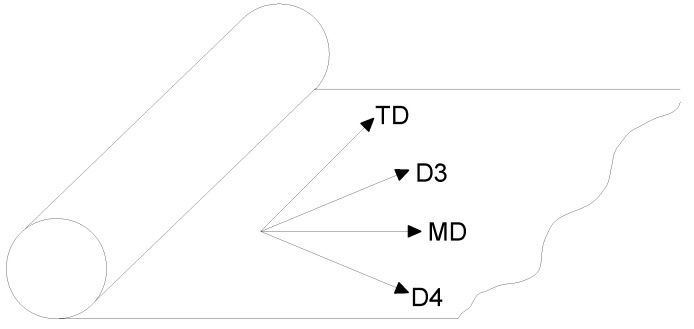
Sketch of the material roll.

**Figure 5 polymers-13-01537-f005:**
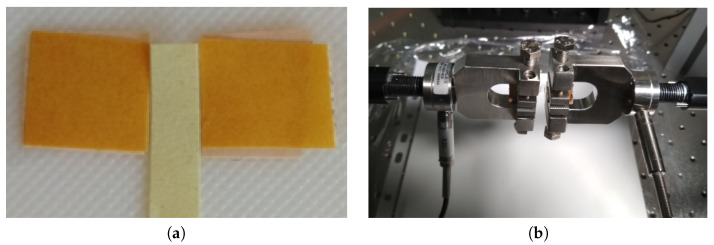
Quasi-static test setup: (**a**) Testing sample of 5 mm × 10 mm. (**b**) Testing jaws.

**Figure 6 polymers-13-01537-f006:**
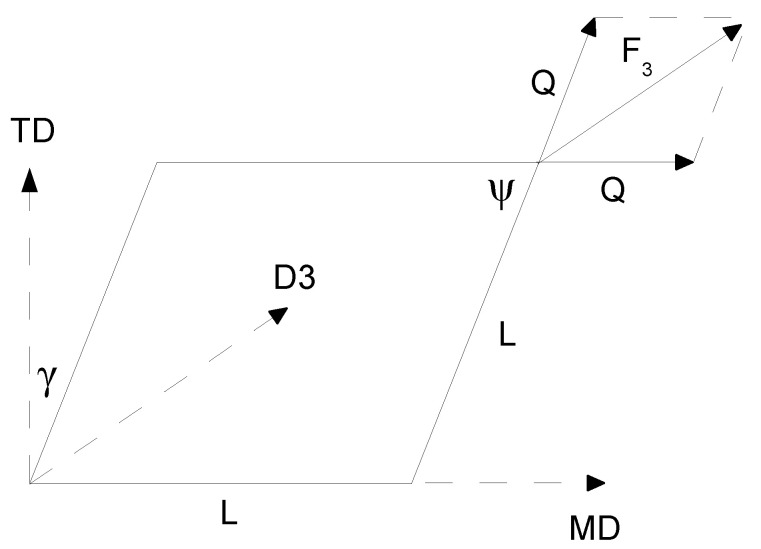
Evaluating the resultant shear stress.

**Figure 7 polymers-13-01537-f007:**
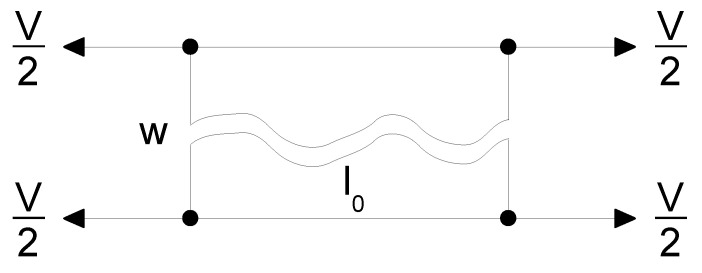
Single-element quasi-static stretching simulation setup.

**Figure 8 polymers-13-01537-f008:**
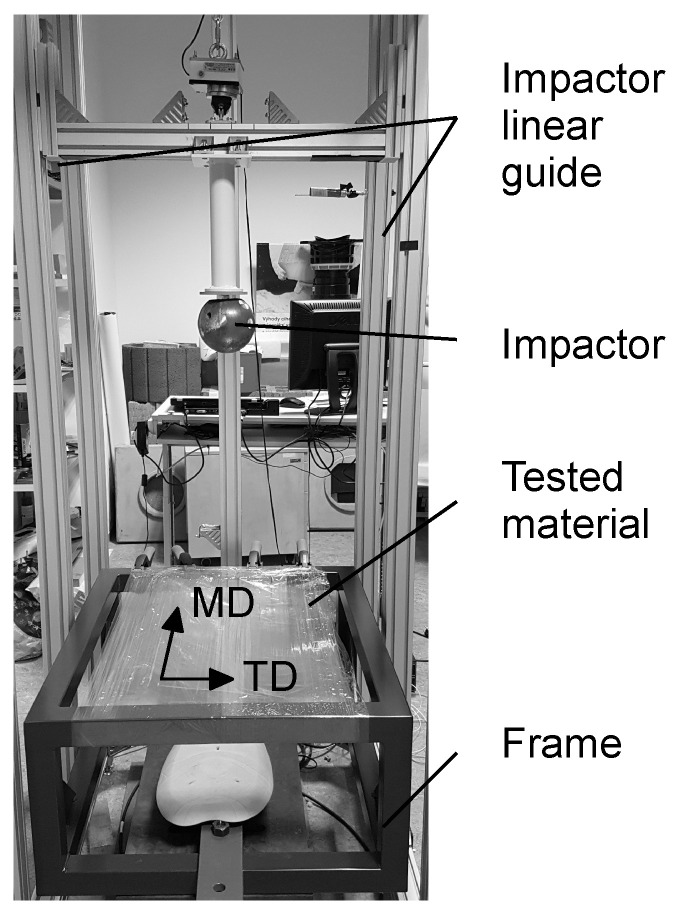
Drop tower.

**Figure 9 polymers-13-01537-f009:**
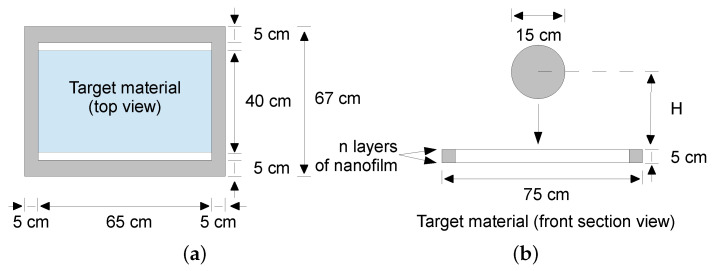
Drop tower scheme: (**a**) Top view. (**b**) Side section.

**Figure 10 polymers-13-01537-f010:**
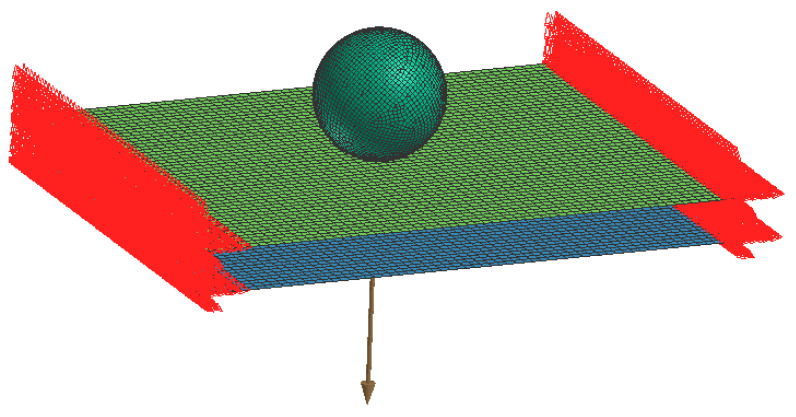
Drop test simulation setup.

**Figure 11 polymers-13-01537-f011:**
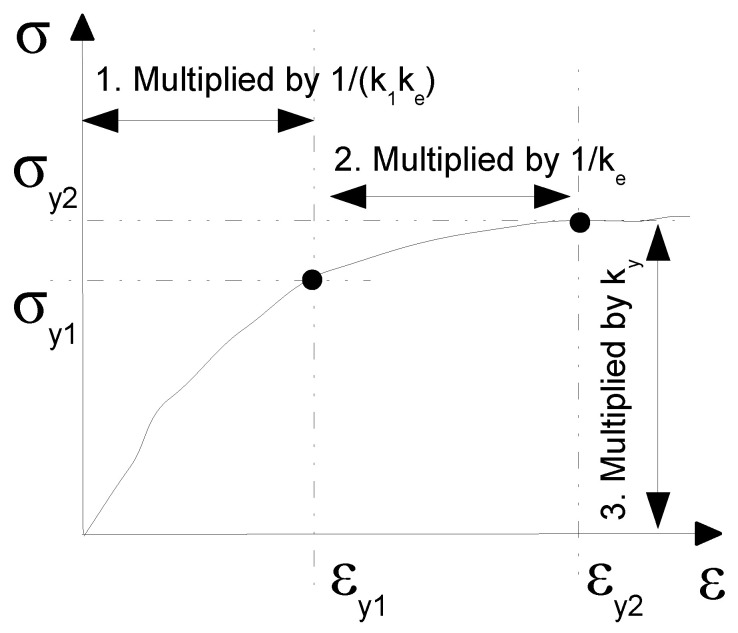
Optimization coefficients in Region I.

**Figure 12 polymers-13-01537-f012:**
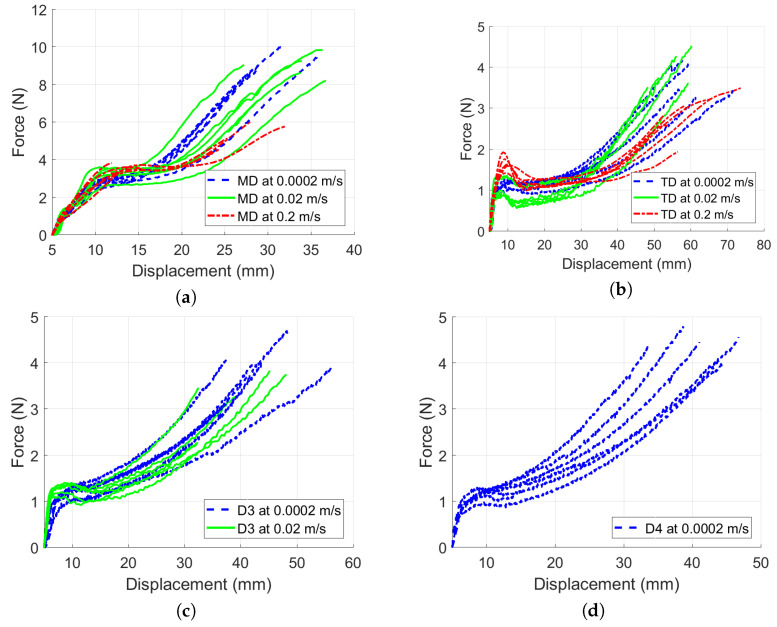
Material response in all directions: (**a**) Force versus displacement in MD. (**b**) Force versus displacement in TD. (**c**) Force versus displacement in D3. (**d**) Force versus displacement in D4.

**Figure 13 polymers-13-01537-f013:**
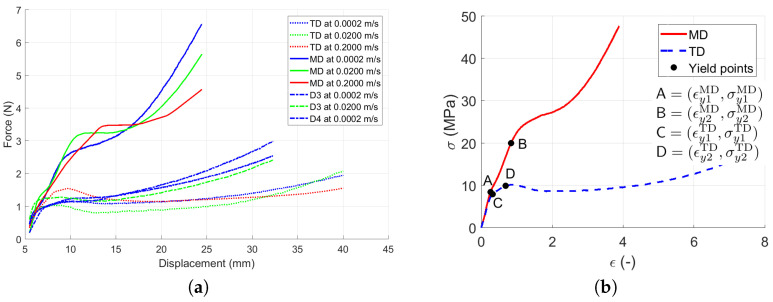
Averaged constitutive material model curves: (**a**) Force versus displacement averaged per direction in all stretching velocities. (**b**) Stress versus strain averaged per direction.

**Figure 14 polymers-13-01537-f014:**
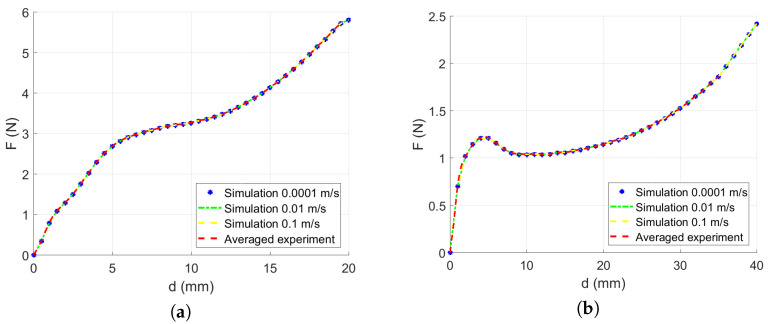
Performance of a single element model: (**a**) Force versus displacement of a single element in the MD. (**b**) Force versus displacement of a single element in TD.

**Figure 15 polymers-13-01537-f015:**
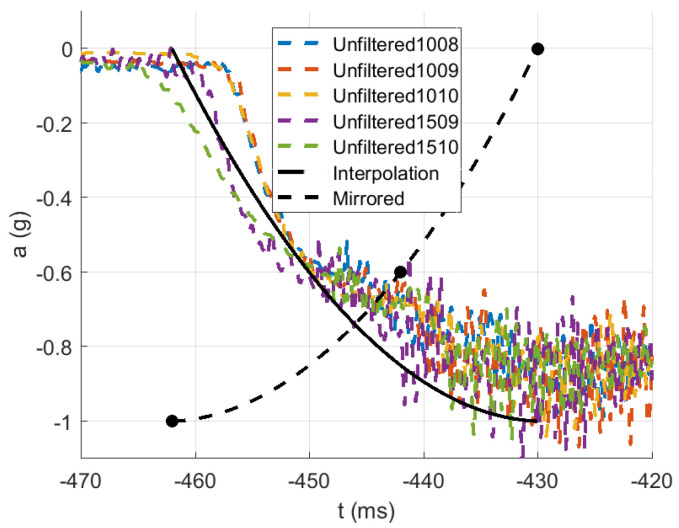
Gravity acceleration ramp.

**Figure 16 polymers-13-01537-f016:**
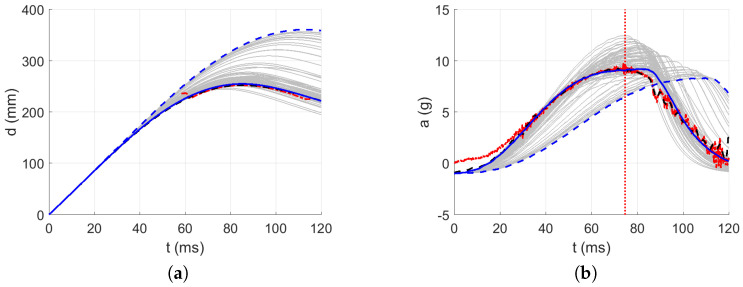
Optimization iterations for the drop height H=10 dm and n=8 layers: (**a**) Impactor displacement. (**b**) Impactor acceleration.

**Figure 17 polymers-13-01537-f017:**
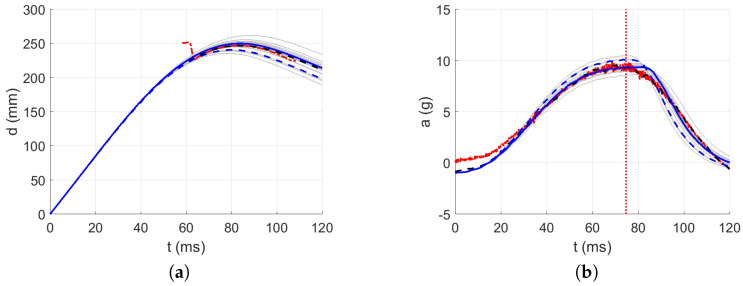
Optimization iterations for the drop height H=10 dm and n=9 layers: (**a**) Impactor displacement. (**b**) Impactor acceleration.

**Figure 18 polymers-13-01537-f018:**
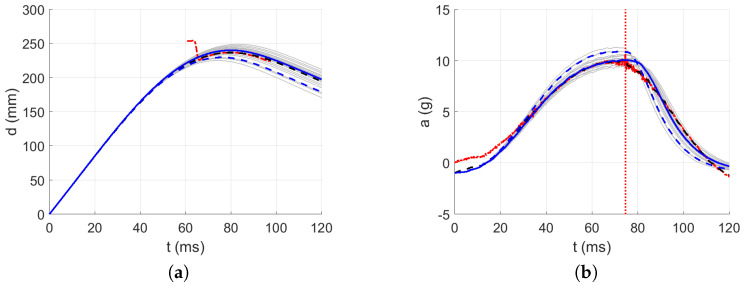
Optimization iterations for the drop height H=10 dm and n=10 layers: (**a**) Impactor displacement. (**b**) Impactor acceleration.

**Figure 19 polymers-13-01537-f019:**
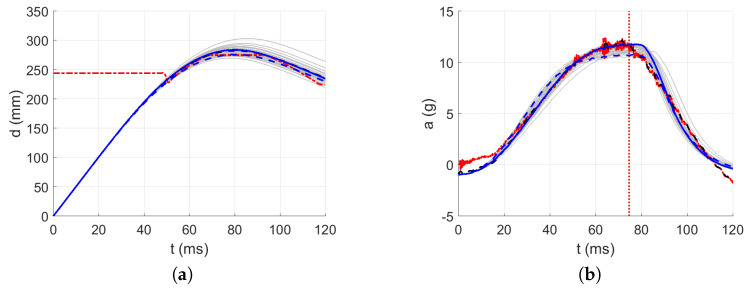
Optimization iterations for the drop height H=15 dm and n=9 layers: (**a**) Impactor displacement. (**b**) Impactor acceleration.

**Figure 20 polymers-13-01537-f020:**
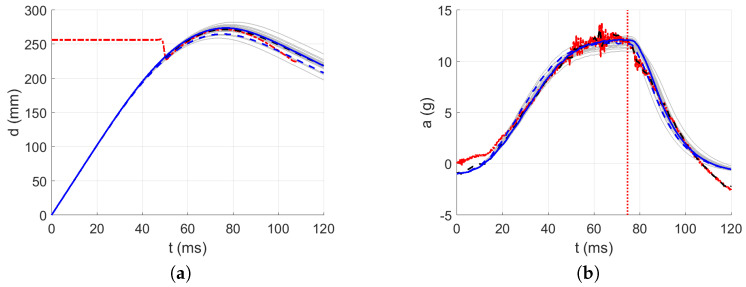
Optimization iterations for the drop height H=15 dm and n=10 layers: (**a**) Impactor displacement. (**b**) Impactor acceleration.

**Figure 21 polymers-13-01537-f021:**
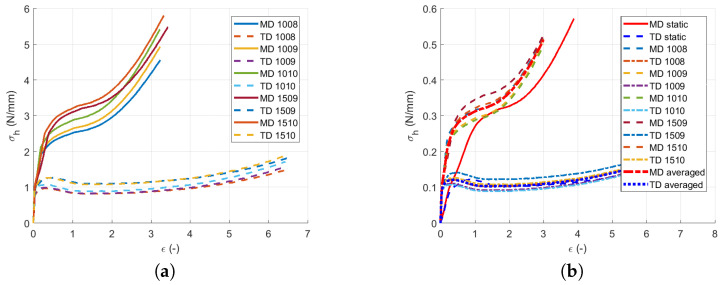
Strain rate-dependent constitutive material model curves: (**a**) Particular drop test response. (**b**) Particular drop test response per layer.

**Figure 22 polymers-13-01537-f022:**
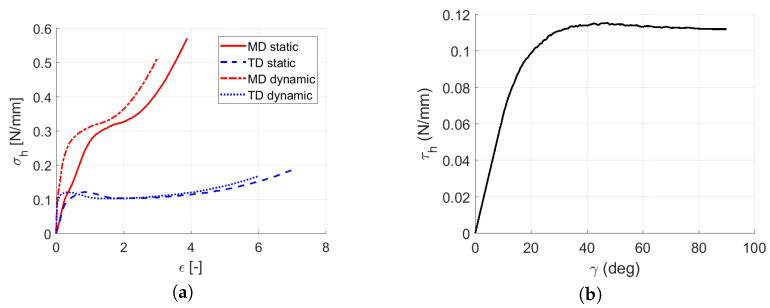
Averaged strain rate-dependent constitutive material model curves: (**a**) Resultant stress versus strain in MD/TD. (**b**) Resultant shear stress versus shear strain.

**Figure 23 polymers-13-01537-f023:**
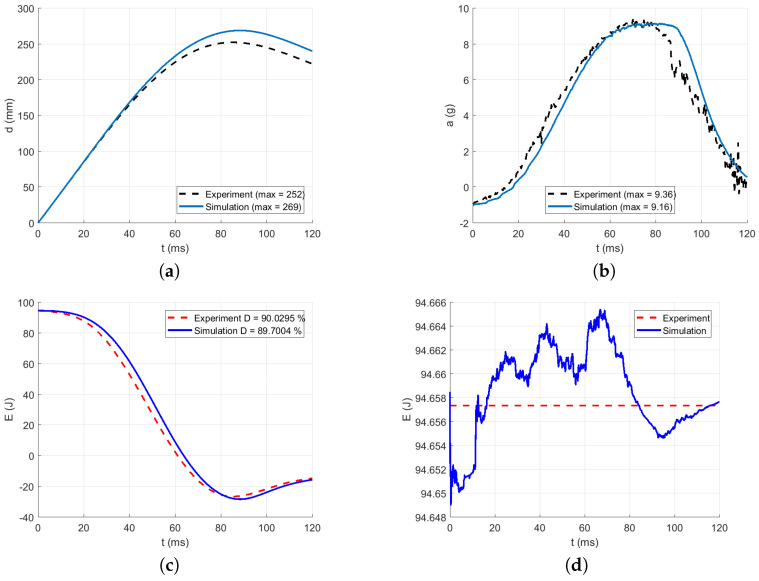
Comparison of the drop test simulation with the experiment for the drop height H=10 dm and n=8 layers: (**a**) Impactor displacement (**b**) Impactor acceleration. (**c**) Impactor energy loss. (**d**) Impactor total energy.

**Figure 24 polymers-13-01537-f024:**
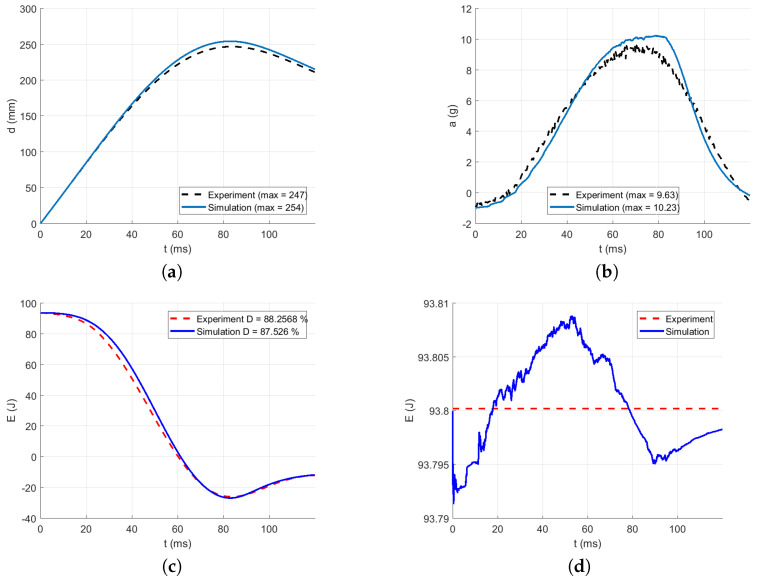
Comparison of the drop test simulation with the experiment for the drop height H=10 dm and n=9 layers: (**a**) Impactor displacement (**b**) Impactor acceleration. (**c**) Impactor energy loss. (**d**) Impactor total energy.

**Figure 25 polymers-13-01537-f025:**
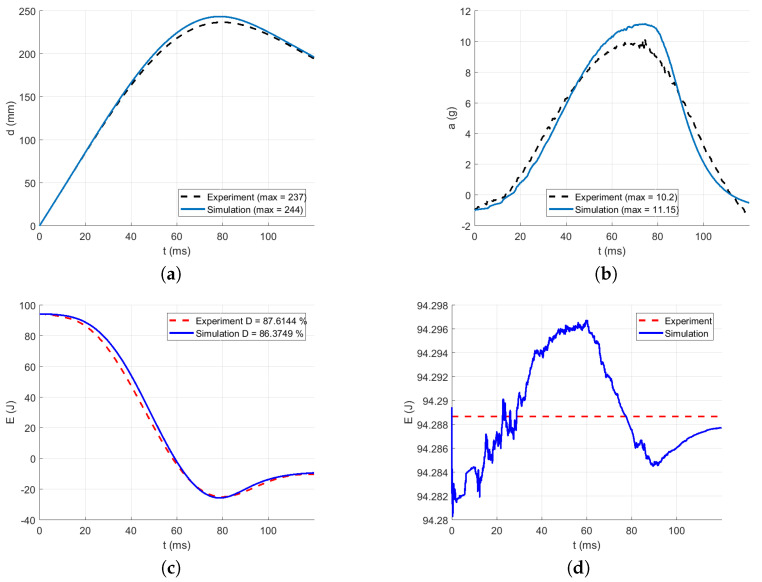
Comparison of the drop test simulation with the experiment for the drop height H=10 dm and n=10 layers: (**a**) Impactor displacement (**b**) Impactor acceleration. (**c**) Impactor energy loss. (**d**) Impactor total energy.

**Figure 26 polymers-13-01537-f026:**
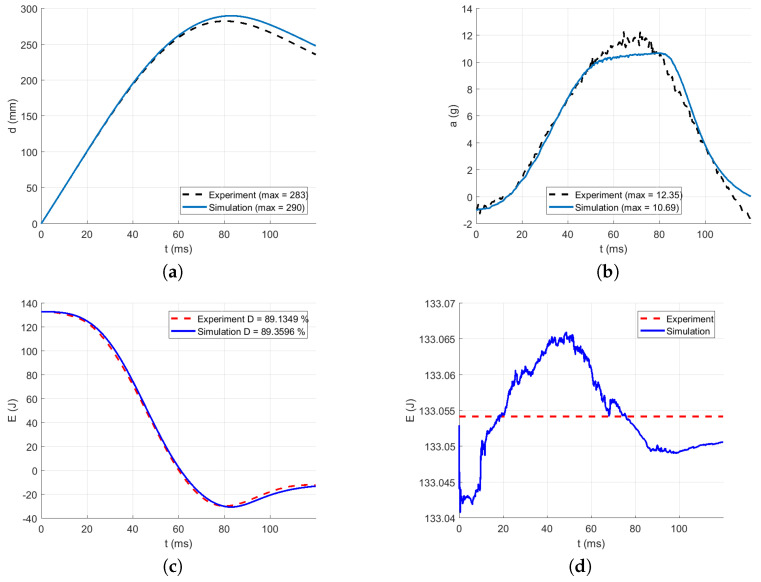
Comparison of the drop test simulation with the experiment for the drop height H=15 dm and n=9 layers: (**a**) Impactor displacement (**b**) Impactor acceleration. (**c**) Impactor energy loss. (**d**) Impactor total energy.

**Figure 27 polymers-13-01537-f027:**
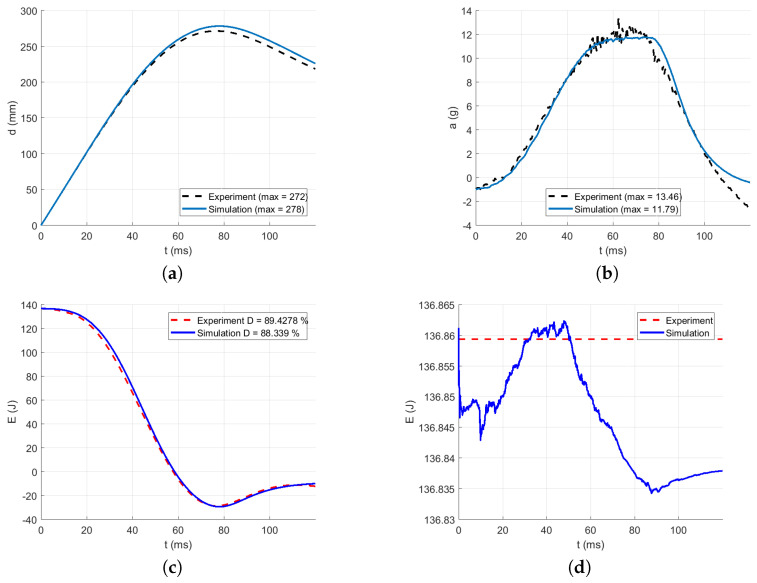
Comparison of the drop test simulation with the experiment for the drop height H=15 dm and n=10 layers: (**a**) Impactor displacement (**b**) Impactor acceleration. (**c**) Impactor energy loss. (**d**) Impactor total energy.

**Table 2 polymers-13-01537-t002:** Quasi-static tests matrix.

Stretching Velocity *v* (m/s)	Direction	Number of Samples *N*
0.0002	MD	6
	TD	6
	D3	6
	D4	6
0.02	MD	6
	TD	6
	D3	6
	D4	6
0.2	MD	6
	TD	6
	D3	6

**Table 3 polymers-13-01537-t003:** Drop tests’ matrix.

Drop Height *H* (dm)	Number of Layers *n*	Optimization	Designation
10	8	√	1008
	9	√	1009
	10	√	1010
15	8	× (material ruptured)	
	9	√	1509
	10	√	1510

**Table 4 polymers-13-01537-t004:** Material properties.

Direction	MD		TD	
**Variable**	**Tensile Stress (MPa)**	**Break Elongation (%)**	**Tensile Stress (MPa)**	**Break Elongation (%)**
Data sheet [22]	29.2	245	14.1	540
Experiment	29.3	139	16.2	701
Error (%)	0.5	−43	15	30

**Table 5 polymers-13-01537-t005:** Yield points ($\sigma_{hy}$ means the resultant yield stress).

Yield Point	MD			TD		
	$\epsilon_{y}$ (-)	$\sigma_{y}$ (MPa)	$\sigma_{\mathrm{hy}}$ (N/mm)	$\epsilon_{y}$ (-)	$\sigma_{y}$ (MPa)	$\sigma_{\mathrm{hy}}$ (N/mm)
1	0.26	8.4	0.1	0.33	8	0.1
2	0.84	20	0.24	0.69	10	0.12

**Table 6 polymers-13-01537-t006:** The Young modulus.

Direction	MD	TD
StretchingVelocity *v* (m/s)	0.0002	0.02	0.2	0.0002	0.02	0.2
Young	44	63	63	67	36	42	54	76	55	41	35	26
modulus	57	76	63	47	37	50	39	53	64	47	31	30
E (MPa)	63	41	81	70	31	30	55	44	49	58	27	38
Young	57	65	38	53	52	31
modulus	53	46
E (MPa)	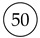

**Table 7 polymers-13-01537-t007:** Energy absorption.

Drop Height *H* (dm)	10	15
Number of layers *n*	8	9	10	9	10
Impact velocity *v* (m/s)	4.16	4.14	4.18	4.94	5.01
Energy absorption *D* (%)	90.03	88.26	87.61	89.13	89.43
Energy absorption D¯ (%)	88.63	89.28
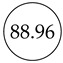

**Table 8 polymers-13-01537-t008:** Optimized coefficients.

Drop Height *H* (dm)	10			15	
Number of layers *n*	8	9	10	9	10
Number of iterations	278	152	205	234	221
First part stiffness multiplier $k_{1}$ (-)	2.75	2.89	2.99	3.14	3.07
Stiffness multiplier $k_{e}$ (-)	3.41	3.47	3.55	1.69	2.29
Yield stress multiplier $k_{y}$ (-)	1.00	0.91	0.88	1.16	1.04
Acceleration error $E_{s}$ (%)	3	3	2	2	3
Displacement error $E_{d}$ (%)	1	1	1	0	1

**Table 9 polymers-13-01537-t009:** Final results.

Drop Height *H* (dm)	10			15	
Number of layers *n*	8	9	10	9	10
Acceleration error $E_{s}$ (%)	6	8	10	6	3
Displacement error $E_{d}$ (%)	5	2	2	3	2

## Data Availability

The data presented in this study are openly available in zenodo at https://doi.org/10.5281/zenodo.4745000.

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
