# Peer review of "Identification of the LLDPE Constitutive Material Model for Energy Absorption in Impact Applications"

_polymers, 2021, doi:10.3390/polym13101537_

Round 1

Reviewer 1 Report

This paper presents an experimental and modelling approach for LLPDE material to be used in absorption energy impact application.

The publication has some merit, however, many issues should be addressed before publication being in the present form not flowable and difficult to read due to the uncompletely reported optimization model and equations.

I would suggest to the authors to address the manuscript according to the following comments, as major revisions need to be carried out.

  • Through all the manuscript, the references should be rewritten according to the journal guidelines. Some reference are reported referring the author name (i.e. pag. 1/21 at line 30 “…patented by Hanuliak (2018)…”), some others indicating the figure of the final reference list (i.e. pag. 2/21 “…compared to LPDE [3]”.
  • Please report in a suitable order the used acronyms (e.g. at pag 2/21 line 57, the “ MD” tear strength is reported but only at page 4/21 this acronym is specified. It would be reasonable to report the abbreviation at the beginning of the manuscript.
  • Figure 3 is not reporting a “Typical stress-strain curve of LLPDE” as indicated in the capture, but rather an "F vs d" curves as indicated on the graph axis. Either change the capture of the figure.
  • 3/21 line 86, the authors report “….LLDPE thin foil produced by Tic (2020)”. This reference is not clear; it is referring to the producer website. Please rewrite this reference clearly according the journal format and add more details about the commercial used LLDPE thin foil.
  • 4/21 line 101, the authors report “….except the v = 0.2 m/s, where D4 was already not necessary to be measured.”. Please explicitly define why the D4 at mentioned  velocity was not experimentally determined.
  • 5/21 Line 103. Please specify the considered value for the gauge length, l0, of the sample as implemented in eq. 1.
  • It is not clear why the author performs the simulation by implementing the model in fig. 7 into the Virtual Performance Solution (VPS by ESI Group), version 2020. Simulation of the tensile quasi-static stretching of the considered samples being this test carried out experimentally.
  • 6/21 line 129. The following sentence is reported “The energy absorption was calculated from the dynamic experimental measurements and other numerical parameters feeding the material model were used as proposed by the VPS manual [24].” It is not clear why the authors added this sentence, I would suggest to add more comments and informations.
  • 6/21 line 140 the authors report “A special drop tower was designed for this purpose, see Figure 8.” Neverthless in the following paragraph (pag. 7/21 line 150) is written that “Due to the design limitations (limited maximum height of the drop tower), only the height H = 1.5 m corresponding to the velocity v0 = 30 km/h was considered.” It is not clear why the designed special drop tower did not take into consideration the necessary height to perform the higher velocity of 50 km/h safety velocity test.
  • Pagg 6-7/21 lines 143-145 the authors report the following two sentences “Typical impacts for testing safety systems are designed for the velocities v0 equal to 30 km/h and 50 km/h…” and “the human head is approximately m = 5 kg..”. I would suggest adding some references to support these sentences
  • I would suggest to the authors to add more comments regarding eq. 10 and 11, in particular it would be necessary to report explicitly the relationship in eq. 11 computing an example being the notation in the line 189-192 not clear.
  • Pag 8/21 The section "dynamic material parameters identification"  is indeed the main computational and optimization procedure of the overall idea of this manuscript however as written is suffer of multiply flaws such as:
    • Equations 12 and 13 should be better referred to the figure 3 being the notation relative to the strain but the figure reporting F vs d.
    • Source of the eq.12 is not clear. The authors are invited to explain the relationship and mainly the geometry factors ki, which are stiffness multiplier.
    • The terms "epsilon1" and "epsilon2" are not clearly identified in the figure, thus reader is confusingly following the implementation and building of the equations .
    • My suggestion is to revised thoroughly page 8/21 and 9/21 allowing the reader to follow the implementation procedure. Please consider to add a final annex reporting all the equationsw.
  • Pag 10/21 , Figure 11 c reports the D3 direction test at only two out of three velocities. Why?
  • Pag 10/21 , Figure 11 c reports the D4 direction test at only one out of two velocities. Why?

Reviewer 2 Report

This paper is on impact loading responses of a linear low density polyethylene. The experimental work is nicely done to show the diversified loading types on the deformation behavior of the polymer. The data support conclusions. Minor changes could be made as shown below:

1. In the second sentence of the Abstract: "packaging material" should be "packaging materials".

2. In the same sentence", the linear low-density polyethylene" may be corrected as ", the linear low-density polyethylene (LLDPE)".

3. In the Figure caption, "Figure 3. Typical stress-strain curve of LLDPE ." may be changed into "Figure 3. Typical stress-strain curves of 
LLDPE with different types of loading." for more accuracy.

4. Citation sources for Figures 1 and 3 may be added into their captions.

5. In reference [25] and [26], the article titles should not be in capital letters for each of the words. 

Round 2

Reviewer 1 Report

The paper was revised taking into account the indicated comments from the first revision stage however still some “gray” issue arise from the definition and balance of eq. 12 and 13.
The meaning and the graphical representation of the terms k1 and stiffness k is not clear. I would rather define differently the constants in eq. 12 and 13 according to their meaning. My suggestion is to devote more to the optimization function being the most interesting section of the whole iterative procedure whose initial point takes the quasi-static experimental curve.
Regarding Figure 12 (b) the graph does not report the indication related to the sigma.
Pease revise the paper to elucidate the optimization procedure as in the present form it is not fully clear. I would suggest to write the optimization function with all the variable term which will be optimised. 

Regarding Figure 12 (b) the graph does not report the indication related to the sigma.

PLease revise the paper to elucidate the optimization procedure as in the present form it is not fully clear. I would suggest to write the optimization function with all the variable term which will be optimised. 

Round 3

Reviewer 1 Report

The paper was revised and the addition of further information and text within the manuscript will surely be of help for the reader. I would like to suggest this paper for publication in the present form